# Thermal Battery for Electric Vehicles: High-Temperature Heating System for Solid Media Based Thermal Energy Storages

## Volker Dreißigacker

German Aerospace Center, Institute of Engineering Thermodynamics, Pfaffenwaldring 38–40,
70569 Stuttgart, Germany; volker.dreissigacker@dlr.de; Tel.: +49-711-6862449; Fax: +49-711-6862747

**Abstract:** Thermal energy storage systems open up high potentials for improvements in efficiency and flexibility for power plant and industrial applications. Transferring such technologies as basis for thermal management concepts in battery-electric vehicles allow alternative ways for heating the interior and avoid range limitations during cold seasons. The idea of such concepts is to generate heat electrically (power-to-heat) parallel of charging the battery, store it efficiently and discharge heat at a defined temperature level. The successful application of such concepts requires two central prerequisites: higher systemic storage densities compared to today's battery-powered PTC heaters as well as high charging and discharging powers. A promising approach for both requirements is based on solids as thermal energy storage. These allow during discharging an efficient heat transfer to the gaseous heat transfer medium (air) due to a wide range of geometric configurations with high specific surfaces and during charging high storage densities due to use of ceramic materials suitable for high operating temperatures. However, for such concepts suitable heating systems with small dimensions are needed, allowing an efficient and homogeneous heat transfer to the solid with high charging powers and high heating temperatures. An appropriate technology for this purpose is based on resistance heating wires integrated inside the channel shaped solids. These promise high storage densities due to operating wire temperature of up to 1300 °C and an efficient heat transport via radiation. Such electrically heated storage systems have been known for a long time for stationary applications, e.g., domestic storage heaters, but are new for mobile applications. For evaluation such concepts with regard to systemic storage and power density as well as to identify preferred configurations extensive investigations are necessary. For this purpose, transient models for the relevant heat transport mechanisms and the whole storage system were created. In order to allow time-efficient simulations studies for such an electrical heated storage system, a novel correlation for the effective radiation coefficient based on the Fourier Number was derived. This coefficient includes radiation effects and thermal conduction resistances and enables through its dimensionless parameterization the investigation of the charging process for a wide range of geometrical configurations. Based on application-typical specifications and the derived Fourier based correlation, extensive variation studies regarding the storage system were performed and evaluated with respect to systemic storage densities, heating wire surface loads and dimensions. For a favored design option selected here, maximum systemic storage densities of 201 Wh/kg at maximum heating wire surface loads of 4.6 W/cm$^2$ are achieved showing significant benefits compared to today's battery powered PTC heaters. Additionally, for proofing and confirming the storage concept, a test rig was erected focusing experimental investigations on the charging process. For a first experimental setup-up including all relevant components, mean temperature-related deviations between the simulative and the experimental results of 4.1% were detected and storage temperatures of up to 870 °C were reached. The systematically performed results confirm the feasibility, high efficiency, thermodynamic synergies with geometric requirements during thermal discharging and the potential of the technology to reach higher systemic storage densities compared to current solutions.

**Keywords:** solid media thermal energy storage system; high systemic storage densities; high temperature heating system; heat supply for electric vehicles

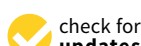



## 1. Introduction

Commitments to the Paris Climate Agreement force global efforts to reduce greenhouse gas emissions in all sectors. Therefore, in addition to the expansion of renewable energy sources and efficiency improvements, increasing electrification of previously fossil-fueled processes is required. Within the transport sector, battery electric vehicles offer a high potential to achieve the targeted goals, but need increased technology developments to improve their attractiveness. In particular, due to the absence of combustion waste heat, alternative thermal management concepts are necessary to avoid range losses of up to 50% [1,2] during the cold season for heating the interior.

Thermal energy storage options [3] extended by electric heating systems are a promising approach facing the challenges ahead, allowing an innovative heat supply instead of today's battery-powered PTC (Positive Temperature Coefficient) heaters. The basic principle is to heat electrically the storage medium parallel of charging the battery, store thermal energy efficiently and to release it at a defined temperature level during vehicle drive. Such thermal storage technologies allow in a time-decoupled operation improvement in systemic flexibility and efficiency and are widely investigated in stationary applications, e.g., industrial [4,5] or power plant [6,7] processes. Depending on the application, media, temperature level or systemic requirements sensible [8,9], latent [10,11] or thermochemically [12,13] based thermal energy storage options are suitable. A transfer of such thermal storage technologies opens up great potentials in the transport sector for future thermal management concepts, but is strongly linked with two crucial requirements: high systemic thermal storage densities and high charging/discharging powers.

So far, several studies have been performed for thermal storage systems based on phase change materials (PCM) for heat supply [14–16], for preheating of catalysts [17], for combustion engines [18] and for cooling applications [19]. New ways for thermal management concepts based on thermochemical energy storage systems (solid/gas reaction) are described in [20], offering an alternative heat and cold supply especially for fuel cell vehicles. In terms of heating demand, each of those thermal storage technologies opens up great potentials to reach high systemic thermal storage densities or high charging or discharging powers, but show limitations for the overall process. A promising solution to fulfil all requirements is given by electrical heated solid media thermal energy storage systems. Such regenerator-type options enable together with efficient thermal insulations high systemic storage densities due to the use of ceramic materials suitable for high operation temperatures and reach high discharging power densities due to high specific heat transfer surfaces as well as a direct contact between the solid and the fluid phase.

Such a storage system with honeycomb-shaped ceramic solids and a bypass concept for controlled discharging outlet temperatures was firstly investigated in [21,22] with vehicle-typical boundary conditions. The results showed–compared to today's battery-powered heating systems as well as to alternative thermal storage systems–high systemic storage densities of up to 160 Wh/kg at constant thermal discharge powers of 5 kW. Additional optimizations with regard to a novel thermal insulation concept were described in [23] and led to further increases of the systemic storage density up to 285 Wh/kg with simultaneously improved operational flexibility. Hereby, one fundamental condition was presupposed in all investigations: a compact and powerful heating system with operating heating temperatures of more than 1000 °C.

Today's commercial high-temperature cartridge heaters use resistance-based metallic heating wires embedded within plate- or cylinder-shaped elements in electrically insulating materials. However, due to efficiency losses through heat conduction and thermal contact resistances, temperatures of only up to 800 °C [24] are reached at the surface of the heating elements, whereby the heating wires themselves operate at temperatures of more than 1000 °C. At the same time, such commercial solutions lead to a significantly increased integration effort within the honeycomb structures and thus to limitations in effective heating power or increased space demands. Alternative electric heating technologies such as inductive processes [25] enable significantly higher operating temperatures if adequate

materials are selected, but are not suited for mobile applications in battery electric vehicles due to their increased complexity, space and electrothermal requirements.

Therefore, to overcome actual limitations, an alternative concept based on commercial heating wire solutions is presented. The basic idea is to integrate the heating wires directly into the electrically non-conductive ceramic honeycomb structure and to heat the solid via thermal radiation. Thus, the heating wire operating temperatures of 1000 °C can be used directly, a homogeneous heating can be achieved by a uniform distribution of the heating wires within the honeycomb, and heating system dimensions can be significantly reduced. Such electrically heated storage systems have been known for a long time for stationary applications, e.g., as domestic storage heaters [26], but are new for mobile applications. Thus, due to the high significance of storage density at BEV for heating the interior and for evaluation of the potential of those thermal storage systems, detailed simulation studies and experimental investigations are needed.

## 2. Modeling

For systematic investigations and identification of efficient design solutions, models of the solid based thermal energy storage system, the heating wire and the surrounding thermal insulation as well as for central heat transport mechanisms are required.

Due to the small dimensions of the storage concept and the associated increased requirements for heat losses–especially at the very high storage temperatures–a computationally efficient 2D porosity model is used for the overall system. Basic relationships to the transient heat balance equations as well as to the heating power term are described in Section 2.1.

A central parameter within the overall 2D model of the storage system regards the effective radiation coefficient between the heating wire and the honeycomb structure including heat transport by radiation and conduction. Its parametrized formulation for a wide range of geometric configurations is of central importance to allow time-efficient simulation studies without losing significant computational accuracy. For this purpose, a detailed model of the individual heat transport mechanisms is presented in Section 2.2 forming the basis for a novel dimensionless parameterization.

### 2.1. Thermal Storage System

The basis for investigations on electrical heating within the storage system are transient models for the solid medium and heating wire. The heat balance equation for the cylindrical honeycomb structure (index *S*) is based on a 2D porosity model [27], while the open heating wire (index *P*) is based on a 0D model due to negligible local temperature gradients. A schematic representation of the thermal energy storage system with the associated relevant variables is shown in Figure 1.

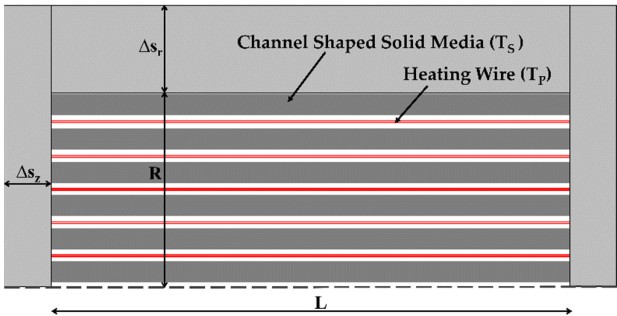

**Figure 1.** Channel shaped solid medium with integrated heating wire and surrounding thermal insulation (convective discharging mode with bypass-configuration not illustrated—see [21]).

Considering the honeycomb void fraction $\varepsilon$ and volume $V$, the axial $z$ and radial $r$ temperature profile $T_S$ over time $t$ is obtained from Equation (1):

$$(1 - \varepsilon)\rho_S c_S \frac{\partial T_S}{\partial t} = \lambda_S \frac{\partial^2 T_S}{\partial z^2} + \lambda_{S,r}\left(\frac{\partial^2 T_S}{\partial r^2} + \frac{1}{r}\frac{\partial T_S}{\partial r}\right) + k_{rad}\frac{O_S}{V} x \left(T_P^4 - T_S^4\right) \tag{1}$$

and the temperature of the heating wire $T_P$ with a volume of $V_P$ at an electrical heating power $P$ according to Equation (2):

$$\rho_P c_P \frac{\partial T_P}{\partial t} = k_{rad}\frac{O_S}{V_P} x \left(T_S^4 - T_P^4\right) + \frac{P}{V_P} \tag{2}$$

Here, the parameter $x$ (heating wire assignment) was implemented to capture the homogeneously distributed number of honeycomb channels with heating wire in relation to the maximum number of channels. The heat transport between the heating wire and the honeycomb surface $O_S$ takes place via the effective radiation coefficient $k_{rad}$, which includes radiation and conduction effects. A model-based determination of this coefficient as a function of the relevant influencing variables is described in Sections 2.2 and 3.1. Both parameters were implemented at the first time inside a porous based approach to allow time-efficient simulations of such a storage system without losing geometrical related information regarding the local structure.

For the density $\rho$, the specific heat capacity $c$ and the thermal conductivities $\lambda$, temperature-averaged material quantities are used. The calculation of the effective thermal conductivity in radial direction ($\lambda_{,r}$) is defined according to [28]. This value considers the effect of void fraction inside the channel shaped honeycomb structure leading to a lower radial heat transport.

In addition to the transient heat balance equations, axial and radial boundary conditions for the honeycomb structure are needed. These include heat losses to the ambient temperature $T_U$ at the axial frontal areas (Equation (3)) at $z = 0$ & $z = L$ and at the shell surface at $r = R$ (Equation (4)) as well as symmetrical boundary conditions at $r = 0$ (Equation (5)).

$$\lambda_S \frac{\partial T_S}{\partial z}\bigg|_{z=0 \, / \, z=L} = k_{Ins,z}\left(T_U - T_S\right) \tag{3}$$

$$\lambda_{S,r} \frac{\partial T_S}{\partial r}\bigg|_{r=R} = k_{Ins,r}\left(T_U - T_S\right) \tag{4}$$

$$\frac{\partial T_S}{\partial r}\bigg|_{r=0} = 0 \tag{5}$$

The stated heat transfer coefficients $k_{Ins,z}$ and $k_{Ins,r}$ include heat conduction through the thermal insulation (index *Ins*) as well as convective heat transfer ($\alpha_U$) to the environment according to [29]. The axial heat transfer coefficient results from Equation (6):

$$\frac{1}{k_{Ins,z}} = \left(\frac{\Delta s_z}{\lambda_{Ins}} + \frac{1}{\alpha_U}\right) \tag{6}$$

and the radial heat transfer coefficient related to the shell surface of the honeycomb structure from Equation (7):

$$\frac{1}{k_{Ins,r}} = \left(\frac{R}{\lambda_{Ins}}\log\frac{R+\Delta s_r}{R} + \frac{1}{\alpha_U}\frac{R}{R+\Delta s_r}\right) \tag{7}$$

The geometry quantities $\Delta s_Z$ and $\Delta s_r$ represent the thermal insulation thicknesses at the frontal areas and at the shell surface of the honeycomb structure.

After using finite-difference-method in space for Equation (1) and the associated boundary Equations (3)–(5), the resulting set of differential algebraic equations including Equation (2) are solved in time with a commercial simulation tool (Matlab).

## 2.2. Electrical Heating Power

For charging the storage system, a heating wire is integrated within the honeycomb channels, which generates a thermal power $P$ as a result of an electric current $I$ at a voltage $U$ (resistance heating). For this, the heating wire is passed several times through the honeycomb channels depending on its geometry and heating wire assignment $x$, whereby a homogeneously distributed arrangement is assumed.

For a straight heating wire with constant cross-sectional area, length $L_P$, specific electrical resistance $\kappa_P$ and a maximum permitted current $I_{max}$, the heating wire diameter $d_P$ is obtained from Equation (8) according to [30]:

$$d_P = 2\sqrt{\frac{I_{max}\,\kappa_P\,L_P}{\pi\,U}} \tag{8}$$

The length of the heating wire $L_P$ can be determined from the heating wire assignment $x$, the maximum number of honeycomb channels $N$ and the honeycomb length $L$ according to Equations (9) and (10):

$$L_P = x\,N\,L \tag{9}$$

$$N = \frac{1}{\varepsilon}\left(\frac{R\,a_V}{2}\right)^2 \tag{10}$$

where the number of channels results from the void fraction $\varepsilon$, the radius $R$ and the specific surface $a_V$ of the honeycomb, defined as the ratio of heat transferring surface to volume ($a_V = O_S/V$).

In addition, for limiting the heating wire surface load a control algorithm is implemented within the model allowing an adjustment of the heating power $P$ to the maximum permitted heating wire temperature $T_{P,max}$ as described in Equation (11):

$$P = min\left[k_{rad}\,O_S\,x\,\left(T_{P,max}^4 - T_{S,max}^4\right), U\,I_{max}\right] \tag{11}$$

Here, the variable $T_{S,max}$ represents the maximum temperature inside the honeycomb structure.

## 2.3. Effective Radiation Coefficient

The electrical heating of the storage system is based on a heating wire passing the honeycomb channels several times and heats the solid via thermal radiation and conduction. A central value for this system is given by the effective radiation coefficient $k_{rad}$, which includes both heat transport mechanism. A consideration of the individual heat transport mechanisms in $k_{rad}$ is necessary, since a detailed geometric representation of the honeycomb structure is not covered by the computationally efficient porosity model. For its determination as a function of the relevant influencing variables, a geometry-resolved model of the honeycomb structure is required serving as a basis for a novel parametrizable formulation of the effective radiation coefficient $k_{rad}$. A schematic representation of the detailed model is shown in Figure 2.

The detailed model–divided into three zones–is based on symmetrical section of a single honeycomb channel with integrated heating wire (zone I), the surrounding solid wall of the honeycomb (zone II) and the honeycomb structure without heating wires (zone III). The corresponding radii of the individual zones can be determined from geometric variables related to the honeycomb structure and the heating wire assignment $x$.

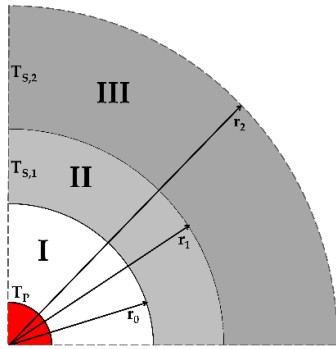

**Figure 2.** Heat transport model: radiation between heating wire and honeycomb (zone I), heat conduction in the surrounding solid wall (zone II) and the porous structure (zone III) of the honeycomb.

For a honeycomb with a void fraction $\varepsilon$ and a specific surface $a_V$, the radius $r_0$ results from Equation (12):

$$r_0 = \frac{2}{a_V}\varepsilon \tag{12}$$

the radius $r_1$ from Equation (13):

$$r_1 = \frac{2}{a_V}\sqrt{\varepsilon} \tag{13}$$

and the radius $r_2$ from Equation (14):

$$r_2 = \frac{2}{a_V}\sqrt{\frac{\varepsilon}{x}} \tag{14}$$

Within the solid honeycomb wall (zone II), the radial transient heat balance is based on Equation (15):

$$\rho_S c_S \frac{\partial T_{S,1}}{\partial t} = \lambda_S \left( \frac{\partial^2 T_{S,1}}{\partial r^2} + \frac{1}{r}\frac{\partial T_{S,1}}{\partial r} \right) \tag{15}$$

and for the surrounding porous honeycomb structure (zone III) on Equation (16):

$$(1-\varepsilon)\rho_S c_S \frac{\partial T_{S,2}}{\partial t} = \lambda_{S,r} \left( \frac{\partial^2 T_{S,2}}{\partial r^2} + \frac{1}{r}\frac{\partial T_{S,2}}{\partial r} \right) \tag{16}$$

The boundary conditions include for the solid honeycomb wall (Zone II) heat transport by radiation at the heating wire temperature $T_P$ according to Equation (17), for the porous honeycomb structure (Zone III) adiabatic conditions according to Equation (18) and for the interface between the two zones heat conduction according to Equation (19).

$$\lambda_S \frac{\partial T_{S,1}}{\partial r}\bigg|_{r=r_0} = C_{rad}\left(T_P^4 - T_{S,1}^4\right) \tag{17}$$

$$\frac{\partial T_{S,2}}{\partial r}\bigg|_{r=r_2} = 0 \tag{18}$$

$$\lambda_S \frac{\partial T_{S,1}}{\partial r}\bigg|_{r=r_1} = \lambda_{S,r} \frac{\partial T_{S,2}}{\partial r}\bigg|_{r=r_1} \tag{19}$$

The pure radiation parameter $C_{rad}$ in Equation (17) is defined for a configuration with concentric cylinders. With regard to the honeycomb surface at $r = r_0$, the radiation parameter $C_{rad}$ is calculated as described in Equation (20):

$$C_{rad} = \sigma \frac{\beta_S\,\beta_P\,\varphi_{SP}}{1-(1-\beta_S)\,(1-\beta_P)\,\varphi_{SP}\,\varphi_{PS}} \tag{20}$$

with $\beta$ as emission coefficients, $\sigma$ as the Stefan–Boltzmann constant, $\varphi_{SP}$ and $\varphi_{PS}$ as geometry-dependent parameters [31] referring to the honeycomb (Index $S$) and heating wire dimension (Index $P$), respectively.

The partial differential Equations (15) and (16) as well as the boundary Equations (17)–(19) are discretized in space by finite-difference-method and the resulting set of differential algebraic equations are solved in time with a commercial simulation tool (Matlab).

Based on the described detailed model for the individual heat transport mechanisms, a parametrized formulation of the effective radiation parameter is determined. Its derivation (see Section 3.1) for a wider range of geometric configurations is new and of central importance to allow time-efficient simulation studies for such a system under investigation. For this purpose, the transient heating process is calculated for both models: the porosity based as described in Section 2.1 and the detail based as described in Section 2.2, whereby $k_{rad}$ is determined in an iterative way by minimizing the temporal (spatial-averaged) temperature deviations. Thereby, adiabatic conditions with $k_{Ins,z} = 0$ and $k_{Ins,r} = 0$ are used for the porosity model to avoid boundary effects.

## 3. Results

Based on the described transient models, systematic investigations for the heat transport and the storage system are performed. Overall goal is to identify geometric configurations promising higher systemic storage densities compared to today's battery-powered PTC heaters and high synergies with geometrical requirements from thermal discharging as describe in [23].

Here, for the storage system under investigation commercially available products suitable for this application were used. These include an $Al_2O_3$ ceramic as solid for the honeycomb structure [32], a ferritic FeCrAl alloy [33] for the heating wire and a microporous material [34] for the thermal insulation. Central temperature-averaged material properties are summarized in Table 1 for the considered operational range.

**Table 1.** Temperature-averaged material properties in a range between $-10\ °C$ to $1000\ °C$.

|  | $\rho$ [kg/m$^3$] | $c$ [J/kgK] | $\lambda$ [W/mK] | $\beta$ | $\kappa$ [$\Omega$ mm$^2$/m] |
|---|---|---|---|---|---|
| Honeycomb (S) | 3991 | 1169 | 11.1 | 0.8 | - |
| Heating Wire (P) | 7250 | 690 | - | 0.7 | 1.4 |
| Thermal Insulation (Ins) | 225 | - | 0.03 | - | - |

For a well-founded evaluation of the storage system, a detailed detection of the heat transport mechanisms is needed. This is done using the effective radiation coefficient $k_{rad}$, which considers the heat transport by radiation and conduction within the overall model in a compact way. Results on this including a novel formulation are presented in Section 3.1. Subsequently, in Section 3.2, systematic investigations for the storage system on the basis of application-typical specifications are performed, depending on the relevant geometric variables. Central contexts to systemic storage densities and heating wire dimensions are explained and storage-optimized results including the selection of a first favored concept are presented.

Mesh studies relating to the averaged honeycomb temperatures were performed in order to reach a high accuracy of the simulation. The investigations showed that with maximum permitted deviations of less than $10^{-4}$ an axial discretization of at least 60 nodes and a radial discretization of at least 30 nodes are required for both simulation models.

### 3.1. Effective Radiation Coefficient

For investigation of the electrically heated storage system, a detailed detection of the heat transport mechanisms inside the effective radiation coefficient $k_{rad}$ is necessary, requiring a geometry-, process- and material-based parametrization. Extensive simulation studies (Table 2) on the relevant influencing factors were performed and $k_{rad}$ was iteratively

determined by minimizing the temperature deviations between the porosity based and the detailed based model. The variations include the solid mass ($m_S$), the length to diameter ratio ($L/D$), the specific surface ($a_V$) and the void fraction ($\varepsilon$) of the honeycomb structure as well as the heating wire assignment ($x$), its diameter ($d_P$) and the charging duration ($\tau$). Here, the heating wire temperature ($T_P$) was set constant at a maximum permitted value of 1000 °C.

**Table 2.** Variation matrix.

| $m_S$ [kg] | $L/D$ [-] | $a_V$ [m²/m³] | $\varepsilon$ [%] | $x$ [%] | $\frac{d_P}{2r_0}$ [%] | $\tau$ [min] |
|---|---|---|---|---|---|---|
| 5–15 | 0.5–4 | 50–600 | 20–80 | 10–90 | 0.05–0.5 | 15–180 |

For an exemplary case, Figure 3 shows the resulting temporal temperature characteristics for four specific surfaces of the honeycomb structure. The squares represent the results based on the detailed model with the pure radiation coefficients $C_{rad}$ and local thermal conduction, the solid lines the results based on the porosity model with the iteratively determined effective radiation coefficients $k_{rad}$. In addition, the corresponding coefficients $C_{rad}$ and $k_{rad}$ are summarized in Table 3.

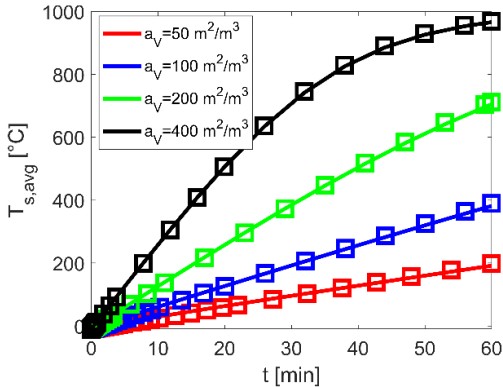

**Figure 3.** Comparison of temporal temperature characteristics: porous model (solid lines) vs. geometry-resolved model (squares) for an exemplary case at $x = 10\%$, $\frac{d_P}{2r_0} = 25\%$, $m_S = 10$ kg, $L/D = 2$, $\varepsilon = 60\%$, $\tau = 60$ min.

**Table 3.** Comparison of the radiation coefficients: iteratively adapted $k_{rad}$ vs. $C_{rad}$.

| $a_V$ [m²/m³] | $C_{rad}$ [W/m²K⁴] | $k_{rad}$ [W/m²K⁴] |
|---|---|---|
| 50 | $7.74 \times 10^{-9}$ | $8.3 \times 10^{-9}$ |
| 100 | $7.9 \times 10^{-9}$ | $8.09 \times 10^{-9}$ |
| 200 | $7.98 \times 10^{-9}$ | $8.02 \times 10^{-9}$ |
| 400 | $8.02 \times 10^{-9}$ | $8.04 \times 10^{-9}$ |

The results show with increasing specific surfaces significant higher temporal temperature characteristics as well as rapprochements between the iteratively determined radiation coefficients $k_{rad}$ to the pure radiation coefficient $C_{rad}$ (Table 3). These results are associated with an increasing heat transport due to two central effects: a total higher number of honeycomb channels with heating wires and decreasing wall thicknesses of the honeycomb structure, thus negligible thermal conduction resistances. Comparable characteristics of different magnitude are also evident in other results within the variation matrix. Additionally, a good agreement in the temporal heating characteristic between the detailed and the porosity model with maximum deviations of lower 5% are reached.

Due to the large number of geometric, process and material configurations, a parameterization of $k_{rad}$ is needed in order to enable a computationally efficient and detailed investigation of the electrically heated storage system throughout the porosity model. For this purpose, an approach based on the Fourier number (*Fo*) according to Equation (21) is assumed

$$Fo = \frac{\lambda_{S,r}}{(1-\varepsilon)\rho_S c_S} \frac{\tau}{l_c^2} \tag{21}$$

whereby the characteristic length $l_c$ as defined in Equation (22) represents through its novel formulation all relevant geometric aspects of such a storage system

$$l_c = r_2 - r_1 = \frac{2}{a_V} \sqrt{\varepsilon} \left( \frac{1}{\sqrt{x}} - 1 \right) \tag{22}$$

The Fourier number as well as the characteristic length defined in this way include all relevant geometric-, process- and material-relating aspects of the honeycomb structure with integrated heating wires. With this dimensionless quantity a comparison between the iteratively determined effective radiation coefficient $k_{rad}$ to the pure radiation coefficient $C_{rad}$ are conducted. Results based on the variation matrix from Table 2 are shown in Figure 4.

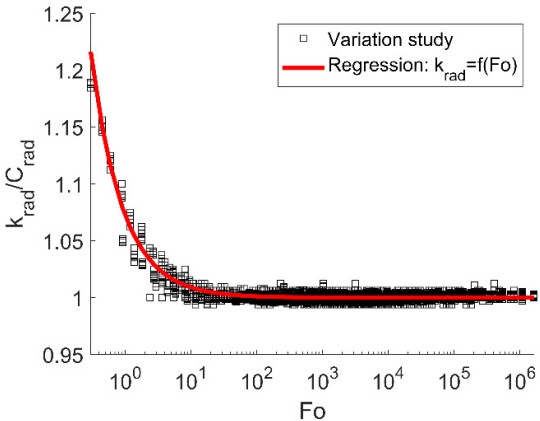

**Figure 4.** Ratio of effective to pure radiation coefficients ($k_{rad}/C_{rad}$) depending on the Fourier number: results from variation study vs. regression function.

The resulting ratios show clearly a correlation between the effective to pure radiation coefficients and pointed out the usability of the novel formulation regarding the characteristic length inside the Fourier number. As can be seen at Fourier numbers $> 10^2$ the heat transport is dominated by thermal radiation without influence of thermal conduction within the honeycomb structure. However, for lower Fourier numbers higher iteratively determined effective radiation coefficients $k_{rad}$ are resulting compared to the pure radiation coefficient $C_{rad}$. This is due to the fact, that the heat transport in the porosity model refers to a homogeneous honeycomb temperature, whereas the detailed model calculates the geometry-resolved temperature distributions in the honeycomb structure and thus captures effects of local thermal resistances. Within the porosity model, this leads especially for thick-walled honeycomb structures or configurations with low heating wire assignments to an underestimation of the heat transport only on the basis of the pure radiation coefficient $C_{rad}$ resulting in higher $k_{rad}/C_{rad}$ ratios at $Fo < 10^2$.

Based on the presented results, it is obvious that a detailed capture of the heat transport mechanisms is achieved by the effective radiation coefficient $k_{rad}$. For implementation inside the porosity model, a power law approach according to Equation (23) is used

$$\frac{k_{rad}}{C_{rad}} = 1 + 0.07276 \, Fo^{-0.903} \tag{23}$$

which allows through its parameterized formulation via the Fourier number a wide consideration of different configurations.

This novel correlation regarding the effective heat transport opens up time-efficient calculations for such electrical heated honeycomb structures without losing accuracy. Therefore, only the pure radiations coefficient must be determined and adapted according Equations (22) and (23) towards the investigated honeycomb-heating-wire configuration.

### 3.2. Thermal Storage System

Based on the models described in Section 2.1 and the parameterized formulation of the effective radiation coefficient according to Section 3.1, systematic investigations of the storage system are performed. Overall aim is to identify storage system configuration with high systemic storage densities complying at the same time restrictions regarding the maximum permitted surface temperature ($T_W$). For this, the maximum thermal insulation thicknesses required to maintain the permitted surface temperature at the frontal areas ($\Delta s_Z$) and at the shell surface ($\Delta s_r$) of the honeycomb structure were iteratively calculated for each design solution. Hereby the extreme case–homogeneously heated honeycomb at the maximum operating heating wire temperature $T_{P,max}$–was assumed.

Application-typical specifications for the storage system with regard to charging duration ($\tau$), storage capacity ($Q$), maximum permitted surface temperature ($T_W$) and electrical power supply ($U$, $I_{max}$) are defined and summarized in Table 4.

**Table 4.** Application-typical specifications as basis for storage system design investigations.

| $\tau$ [min] | $Q$ [kWh] | $T_W$ [°C] | $U$ [V] | $I_{max}$ [A] |
|---|---|---|---|---|
| 30 | 2.5 | 60 | 400 | 16 |

Here, values of 1000 °C and −10 °C were specified for the maximum permitted heating wire temperature ($T_{P,max}$) and for the ambient temperature ($T_U$). The material properties of the heating wire, the honeycomb structure and the thermal insulation were assumed to be constant and can be found in Table 1.

In order to examine the storage system for its performance and systemic storage densities, variation studies were performed on the relevant influencing variables. These include the heating wire assignment ($x$), the specific surface ($a_V$) and the void fraction ($\varepsilon$). Hereby, the minimum storage mass ($m_S$) of the honeycomb structure was iteratively determined for each design solution fulfilling the required specifications as shown in Table 4. Due to neglectable influences of the length-to-diameter ratio on the results within the simulation studies a constant value of $L/D = 2$ was set.

For explanation of the central effects with regard to systemic storage density and heating wire dimensions, results on specific surface and heating wire assignment are presented firstly in Section 3.2.1. Based on these, storage-optimized results are derived in Section 3.2.2 and the impact of void fractions are discussed. Finally, central characteristics of the transient heating process for a favored design option and the resulting storage dimensions are presented in Section 3.2.3.

### 3.2.1. Fundamental Contexts to Systemic Storage Densities

For investigating of such a concept on its systemic storage densities, results depending on the heating wire assignment and the specific surface for an exemplary selected solution set with a void fraction of 42.5% are shown in Figure 5. Hereby the gravimetric systemic storage density ($Q/m_{tot}$) includes the mass of the honeycomb, the heating wire and the thermal insulation and the volumetric systemic storage density ($Q/V_{tot}$) includes the volume of the honeycomb and the thermal insulation.

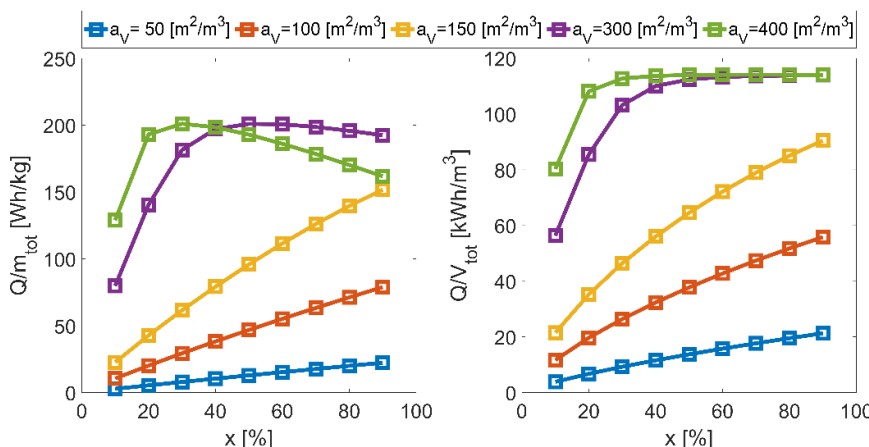

**Figure 5.** Gravimetric and volumetric systemic storage densities depending on heating wire assignment ($x$) and specific surface ($a_V$) at an exemplary void fraction ($\varepsilon$) of 42.5%.

The results show an increase in systemic storage densities up to a maximum value with increasing heating wire assignments at constant specific surfaces. This is due to the fact, that with this associated higher storage temperatures allow decreasing honeycomb masses or volumes, but increase simultaneously the heating wire and thermal insulation dimensions. Specifically, for efficient design solutions with high specific surfaces and hence the lowest honeycomb dimensions, this contrary effect leads to the observed maximum in gravimetric and the plateau in volumetric systemic storage density.

The lower systemic storage densities with decreasing specific surfaces at constant heating wire assignments results from their corresponding smaller number of honeycomb channels. This leads–apart from increasing thermal conduction resistances–to shorter total heating wire lengths ($L_P$) and so to a moderate heating of the honeycomb structure and thus to higher storage masses.

This context is illustrated in Figure 6 showing the systemic storage densities as a function of the resulting heating wire dimensions.

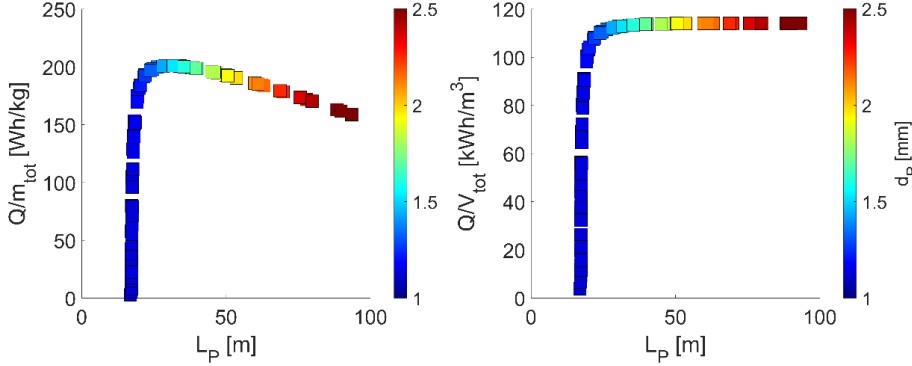

**Figure 6.** Gravimetric, volumetric systemic storage densities and heating wire diameter ($d_P$) depending on heating wire lengths ($L_P$) at an exemplary void fraction ($\varepsilon$) of 42.5%.

The results show a direct correlation between the systemic storage densities and the heating wire dimensions. With increasing heating wire length, a maximum value in the systemic storage densities is reached limited by above-described contrary effect between honeycomb and heating wire or thermal insulation dimensions. Optimal solutions–here with heating wire length of about 30 m–can be directly derived from configurations with appropriate specific surfaces and heating wire assignments. Additionally, the results show correlating heating wire lengths to heating wire diameters ($d_P$) as illustrated in colored dots. This behavior is based on the specified electrical power supply in Table 4 forcing

a constant electrical resistance and thus increasing heating wire diameters with heating wire lengths.

For the results presented here with an exemplary selected void fraction, maximum gravimetric and volumetric systemic storage densities of about 201 Wh/kg and 113 kWh/m$^3$ are reached, whereby these are associated with design solutions for specific surfaces and heating wire assignments. In order to identify only such systems with maximum possible storage densities, iterative optimization simulation studies were performed. The corresponding results with this as well as the influence of void fraction are presented in the following.

### 3.2.2. Optimized Systems with Maximum Gravimetric Storage Density

As explained in Section 3.2.1, maximum possible storage densities are reached based on appropriate configurations regarding the specific surface and the heating wire assignment, whereby the void fraction ($\varepsilon$) was kept constant so far. For a holistic investigation of the storage system, additional variation studies are performed on this. Hereby, the presented results refer to the maximum possible gravimetric storage densities determined through iterative optimization simulations to the heating wire assignment depending on the specific surface. Central results regarding the iteratively optimized systemic storage densities are illustrated in Figure 7.

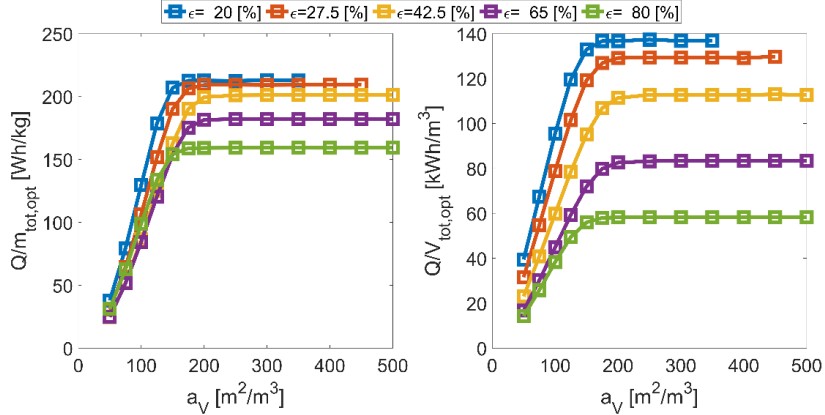

**Figure 7.** Maximum possible gravimetric and associated volumetric systemic storage densities depending on specific surface ($a_V$) and void fraction ($\varepsilon$).

The results show increasing maximum possible systemic storage densities with higher specific surfaces. This is caused by the already mentioned effect regarding the number of honeycomb channels, the corresponding heating wire lengths and thus only a moderate temperature elevation. For those solutions, the iteratively determined results force a maximum heating wire assignment of 100% to reach their maximum possible systemic storage densities. In contrast, at high specific surfaces of 500 m$^2$/m$^3$–thus a maximum temperature elevation of the honeycomb structure–only heating wire assignments of less than 20% are needed.

Additionally, the results point out that for maximum possible storage densities a specific surface of above 200 m$^2$/m$^3$ is required, whereby its magnitude only depends on the void fraction. But with higher void fractions increasing honeycomb and proportionally significantly increasing thermal insulation volumes are associated leading to reductions in systemic storage densities, especially for the volumetric. Furthermore, the results show an absence of solutions at high specific surfaces and small void fractions. This is caused by associated insufficient low channel diameter of the honeycombs compared to the heating wire diameter.

The presented results and contexts point out the significance of high specific surfaces as central prerequisite for maximum systemic storage densities. However, for a holistic evaluation of such systems, additional statements regarding the heating wire surface load

($\dot{q}$) and thus the lifetime are necessary. Therefore, the maximum heating wire surface loads occurring in the transient process are shown in Figure 8 based on the iteratively optimized variation studies.

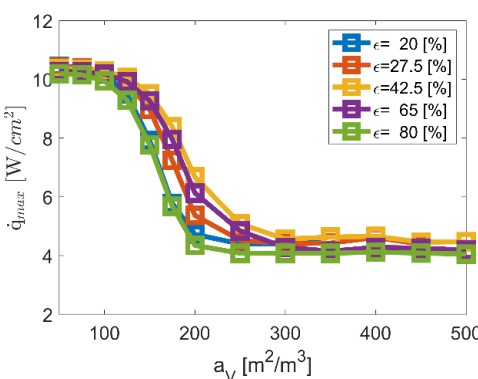

**Figure 8.** Maximum heating wire surface load depending on specific surface ($a_V$) and void fraction ($\varepsilon$).

The results show decreasing maximum heating wire surface loads with higher specific surfaces. This is due to the fact of increasing heating wire diameters and lengths with slight void fraction-related dependencies. However, it is visible that well-suited heating wire surface loads of less than 5 W/cm$^2$ [35] are reached with sufficient high specific surfaces of more than 250 m$^2$/m$^3$ minimizing lifetime-critical conditions in addition to the already moderate heating wire temperature of 1000 °C.

Based on these results for optimized systemic storage densities and corresponding maximum heating wire surface loads, a favored design option is selected meeting also geometrical requirements in terms of specific surface and void fraction for an efficient thermal discharging operation with low pressure drops as describe in [23]. For this, transient characteristics as well as central storage dimensions are presented in the following.

### 3.2.3. Selected Design Option: Temporal Characteristics and Dimensions

Within the wide solution spectrum, a favored design option was selected considering systemic storage densities, maximum heating wire surface loads and thermal discharging requirements. The selected solution is based on a honeycomb structure with a specific surface of 350 m$^2$/m$^3$, a void fraction of 42.5% and a heating wire assignment of 38.4%. Central transient characteristics are shown in Figure 9.

As can be seen, a constant electrical heating power is achieved over a period of about 20 min, whereby a linear temperature elevation of the honeycomb structure from −10 °C to 800 °C or a thermal charging state of 80% is reached. From this point on, the implemented control algorithm leads to a reduction of the heating power to limit the heating wire temperature at 1000 °C. The heating wire itself shows significantly higher temperature gradients at the beginning of the loading process due to lower thermal masses compared to the honeycomb. However, from a heating wire temperature of about 800 °C the dominating radiation heat transport leads to substantial increasing honeycomb temperatures and thus to reduced heating wire temperature gradients.

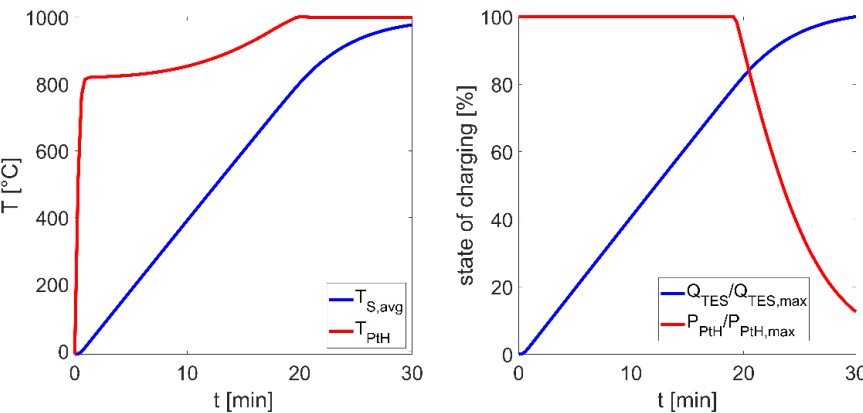

**Figure 9.** Temporal characteristic of a favoured design option: heating wire and (spatial-averaged) honeycomb temperature as well as the normalized electrical heating power and thermal charging state.

For the exemplary favored design option presented here, systemic storage densities of 201 Wh/kg and 113 kWh/m$^3$ with a maximum heating power of 6.4 kW are reached. During charging maximum heating wire surface loads of 4.6 W/cm$^2$ occur at heating wire length and diameters of 30 m and 1.5 mm, respectively. Despite the high storage temperatures, only maximum heat losses of about 76 W occur resulting from the restrictive specification regarding the maximum permitted surface temperature of 60 °C (Table 4) and the small storage dimensions. Central specifications of the selected design option are summarized in Table 5.

**Table 5.** Central specifications of the selected design option.

|  | Honeycomb (S) | Thermal Insulation (Ins) | Heating Wire (P) |
|---|---|---|---|
| *m* [kg] | 7.8 | 4.2 | 0.4 |
| *V* [l] | 3.4 | 18.8 | |

The systematically prepared results regarding systemic storage densities and heating wire surface loads confirm the feasibility and efficiency of such storage systems for the heat supply in BEV. Comparable results are also reached for different electrical power supplies and for charging durations of less than 30 min.

The contexts presented here elucidate the need of high specific surfaces of solid media thermal energy storages in addition with adequate selected void fractions. Comparable geometric conditions are also required during thermal discharging, in order to allow efficient convective heat transport to the heat transfer medium (air). The associated thermodynamic synergies in the cyclic storage process, the achievable systemic storage densities and the commercial availability of such ceramic honeycomb structures confirm the potential of the technology as alternative thermal management concept of battery electric vehicles.

For proof of concept a test rig was erected, enabling due its high flexibility in terms of electrical voltage, heating power and exchangeable storage media wide experimental investigations. First results of an exemplarily honeycomb-heating-wire-configuration were performed and used for validation of the described porous simulation model.

## 4. Proof of Concept: First Experimental Validation

For proofing of the storage concept and for validation, a test rig was erected focusing preliminary experimental investigations on the charging process and statements regarding the quality of the porous simulation model. Central components within the test rig include the honeycomb with integrated heating wires and the thermal insulation. Through its high flexibilities in terms of electrical voltage (10–230 V), electrical current (maximum 16 A) and

exchangeable storage media, wide experimental investigations of different honeycomb-heating-wire-configurations are enabled. The test rig includes all essential components and requirements for the storage concept, allows a validation of central characteristics of the porous simulation model and the identification of critical design aspects for the futural target scale.

For the first experimental setup-up, a ceramic honeycomb (alumina porcelain) with a mass of 6.8 kg, a thermal insulation (calcium silicate) with an insulation thickness of 100 mm and a ferritic FeCrAl based heating wire were [33] selected. The honeycomb has a square cross-section area of 150 × 150 mm, a length of 300 mm and a specific surface of 273.4 $m^2/m^3$ with a void fraction of 62.2%, the heating wire a diameter of 1 mm and a total length of 27.3 m. The heating wire was uniformly integrated in the honeycomb structure, resulting in a heating wire assignment of 43.2%. Within the honeycomb, two thermocouple sets were installed each triangularly at a distance of 100 mm from the frontal areas and placed close to the honeycomb wall. Central aspects of the experimental setup-up can be seen in Figure 10.

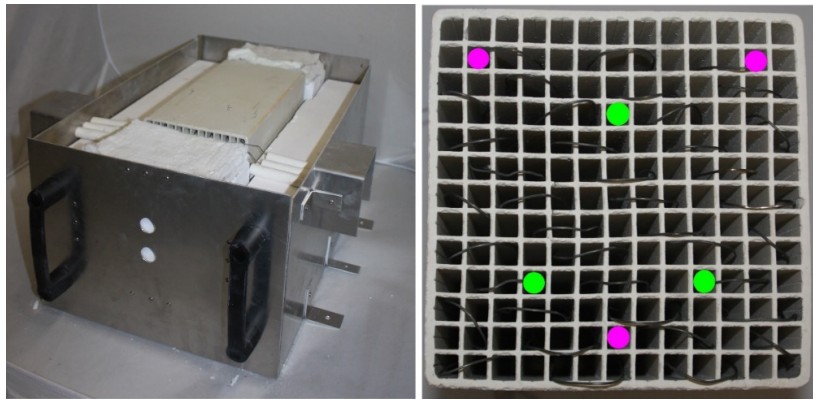

**Figure 10.** Test rig for the experimental investigation of the charging process: honeycomb and thermal insulation (**left**); thermocouple sets ($T_{1-3}$ green colored, $T_{4-6}$ magenta colored) and heating wires (**right**).

First experimental tests on the shown setup-up were performed with an electrical heating power of 0.7 kW at a voltage of 180 V. The transient charging process with the associated experimental and calculated temperatures based on the model explained in Sections 2.1 and 2.2 are shown in Figure 11.

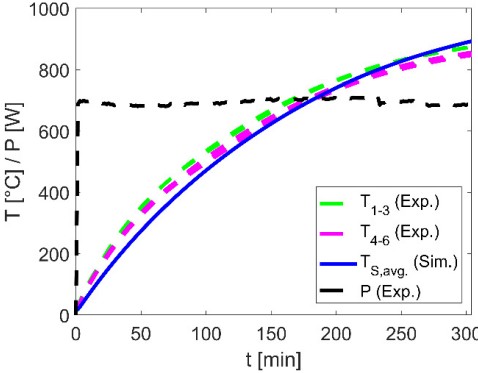

**Figure 11.** Validation: experimental (Exp.) vs. simulation (Sim.) results.

The experimental results show a homogeneous heating despite slightly lower temperatures of the thermocouples ($T_{4-6}$) located closer at the edge region compared to the thermocouples located inside the core region ($T_{1-3}$) of the honeycomb. This is caused due to the thick-walled thermal insulation leading to low heat losses and thus to low

local temperature differences within the honeycomb. Analogous to the experiment, the local temperatures resulting from the porous 2D simulation model showed only slight differences of less than 30 °C. In comparison between the experimental and the averaged simulative temperatures ($T_{S, avg.}$), a good agreement in the temporal heating process over the whole charging period is visible. For the first experimental setup-up with a honeycomb structure close to the specifications of the favored design option as described in Section 3.2.3, maximum temperature-related deviations of 11.1% and mean deviations of 4.1% are determined.

The first experimental investigations on the charging process confirm a good agreement between experimental and simulation results with regard to the temperature characteristics of the honeycomb and a homogeneous heating of the whole storage medium. Thus, the computational efficient porous model including the novel correlation of the effective radiation coefficient $k_{rad}$ are seen as a solid basis for the simulation studies performed in the context of this publication.

## 5. Conclusions

New paths for heat supply in battery-electric vehicles (BEV) are enabled by thermal energy storage systems leading to an increased effective range through reduced battery consumption. However, the successful use of such thermal energy storage systems requires two central prerequisites: high systemic storage densities compared to today's battery-powered PTC heaters and simultaneously high charging and discharging powers. A promising concept to fulfill all requirements is based on solid media as thermal energy storage, but needs technological development in terms of a suitable high temperature electric heating system. For this purpose, a resistance-based heating wires system was investigated offering powerful and efficient charging through thermal radiation at high operating temperatures of up to 1300 °C and a homogeneous heating.

In order to evaluate such concepts as central elements for the heat supply in BEV and to identify favored configurations, detailed models for the heat transport as well as for the storage system were created. For time-efficient simulations studies, a novel Fourier Number based correlation for the effective radiation coefficient including radiation and thermal conduction effects was derived. Through its dimensionless parameterization, holistic investigations of the charging process for a wide range of geometrical configurations were enabled. Systemic results obtained from this showed that high specific surfaces of the honeycomb structure of more than 250 m²/m³ are necessary for high systemic storage densities. Simultaneously, those geometrical configurations lead to low maximum heating wire surface loads of less than 5 W/cm². Additionally, an optimum in the gravimetric systemic storage density is achieved as a function of the heating wire assignment, whereby the honeycomb void fraction only affects its magnitude. Comparable requirements regarding specific surface and void fraction were also identified in investigations performed for the thermal discharging process, through which the potential of the technology even in the cyclic storage operation is confirmed.

For a favored design option selected here, maximum systemic storage densities of 201 Wh/kg with a charging power of 6.4 kW were achieved by simultaneous low maximum heat losses of 76 W for a permitted exterior surface temperature of 60 °C. Comparing the thermal storage densities with those of battery powered PTC heaters operating in a range from 100 Wh/kg to 180 Wh/kg [15,36] for today's commercial Li-Ion systems, the vehicle-systemic and even cost potentials are obviously in spite of drawbacks regarding exergetic potentials. For proofing and confirming the storage concept, a test rig was erected focusing experimental investigations on the charging process and model validations. For a first experimental setup-up including all relevant components of such an electrical heated system, maximum temperature-related deviations between the simulative and the experimental results of 11.1% and mean deviations of 4.1% were determined, whereas storage temperatures of up to 870 °C were reached.

The systematically performed results confirm the feasibility, high efficiency, and thermodynamic synergies with geometric requirements during thermal discharging. Further improvements in systemic storage densities up to 285 Wh/kg are reached by an alternative thermal insulation concept as described in [23] and by higher heating wire temperatures. Additional simulation studies with different electrical power supply up to 400 V and 32 A also showed a large range of solutions with comparably high storage densities. Beyond this, the system enables a high degree of operational flexibility: besides as thermal energy storage system, the presented concept can also be used directly for continuous heat supply, analogous to today's PTC heaters. With its hybrid operational character and its very good scalability in terms of capacity, temperature level and power, the technology offers a wide range of application scenarios for different types of BEV. These results together with environmentally uncritical as well as potentially low-cost materials strengthen the thermal storage technology as an alternative option for the heat supply of BEV and open up the path for a prototypical implementation.

**Funding:** This research received no external funding.

**Conflicts of Interest:** The author declares no conflict of interest.

## Nomenclature

| | |
|---|---|
| R | Radius [m] |
| D | Diameter [m] |
| L | Length [m] |
| $\Delta s$ | Insulation thickness [m] |
| V | Volume [$m^3$] |
| O | Surface [$m^2$] |
| x | Heating wire assignment |
| m | Mass [kg] |
| $a_V$ | Specific surface [$m^2/m^3$] |
| $\varepsilon$ | Void fraction |
| d | Diameter [m] |
| $\alpha$ | Heat transport coefficient [$W/m^2K$] |
| k | Total heat transfer coefficient [$W/m^2K$] |
| t | Time [s] |
| z | Axial dimension [m] |
| r | Radial dimension [m] |
| $\rho$ | Density [$kg/m^3$] |
| c | Specific heat capacity [J/kgK] |
| $\lambda$ | Heat conduction [W/mK] |
| $\kappa$ | Specific electrical resistance [$\Omega\,mm^2/m$] |
| $\beta$ | Emission coefficient |
| T | Temperature [K] |
| Q | Thermal energy [J] |
| $\dot{q}$ | Heating wire surface load [$W/m^2$] |
| $\tau$ | Duration [s] |
| $C_{Rad}$ | Radiation parameter [$W/m^2K^4$] |
| P | Electrical heating power [W] |
| I | Electric current [A] |
| U | Electric voltage [V] |
| Fo | Fourier number |
| Subscripts | |
| S | Solid |
| P | Heating wire |
| Ins | Thermal insulation |
| U | Environment |
| W | Exterior surface |

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
