# Peer review of "Thermal Battery for Electric Vehicles: High-Temperature Heating System for Solid Media Based Thermal Energy Storages"

_applsci, doi:10.3390/app112110500_

Round 1
Reviewer 1 Report
1) The Abstract is too general and only descriptive. In the Abstract the Author should add some of the most important results obtained in this research (its exact values), which cannot be found in other literature. Such addition will highlight the novelty of the presented paper already in the Abstract – at the moment from the abstract cannot be seen any novel elements which are obtained in this paper.
2) In the paper should be added a Nomenclature inside which will be listed and explained all abbreviations, symbols and markings used throughout the paper text and in the equations. Without a Nomenclature it is sometimes hard to track a paper text without constantly turning back to find some meanings. In the Nomenclature will be listed and explained in one place all abbreviations, symbols and markings used throughout the paper text which will surely improve reading experience.
3) In Section 2 – Modeling, should be better explained and highlighted novel elements in the numerical model obtained by the Author. From the presented equations for thermal storage system and for heat transfer, I cannot observe any novel elements which cannot already be found in the available literature. As the whole paper is based on the numerical modelling, in the numerical model are surely expected some novelties, which I cannot see at the moment.
4) The English is understandable, but it should be improved in several sentences or whole paper parts. Many sentences are too long and should be shortened. Please, perform a careful check of the English throughout the paper text.
5) As the Abstract, the Conclusions section should also be improved with the most important obtained results (its exact values). Also the Conclusions seem to be too descriptive and general.
6) The paper is only a theoretical one. I have no doubts that performed calculations are correct (according to adopted assumptions), but in the paper cannot be found any evidence that such energy storage systems can operate as presented or how accurate and precise are the presented results. Measurements of such presented energy storage system during exploitation or in the laboratory are completely missing.
7) As the validation of such system did not exist (direct comparison of calculated and measured results), all the results presented in this paper are at least debatable (or possibly wrong, at least some of them). The accuracy and precision of numerical model are unknown, and there is no evidence that numerical model can track operating parameters of the presented energy storage system during its operation.
8) In relation to numerical modelling – I have never seen such numerical model which can track any real system without its improvements and modifications. At least, not with acceptable accuracy and precision. So, in my opinion, presentation of numerical model and its results can be interesting, but without comparison with real systems, obtained results can show only trends, not an operation of a real system.
9) This paper represents interesting idea, but only the idea, without any connection with system real operation and its applicability cannot be considered as proper and complete scientific research. At least, not according to my opinion.
10) Finally, this paper shows new idea, but without any exact evidences, scientific novelty is non-existing and importance to the scientific community is at least doubtful.
Unfortunately, I cannot recommend publication of this paper.
Reviewer 2 Report
The author present an interesing paper about thermal battery for electric vehicles, where transient models for the relevant heat transport mechanisms and the storage system have been evaluated.
The paper is well written and structured. The performed simulations and the obtained result, together the conclusions are quite relevant n this scientific field.
Author Response
Dear reviewer,
Thank you for your positive feedback. I am very pleased that you consider the work to be worth for publishing.
Some information for you:
- Additions were integrated focusing preliminary experimental validations (new section)
- Novel aspects regarding the simulation tool were highlighted (parameterized effective radiation)
- Additional aspects regarding the flexibility of the storage system (implemented inside the conclusion section)
- All supplements were added in color
Reviewer 3 Report
The issues related to electric car cabin heating are topical and interesting.
One of the solutions currently used are high-voltage electric heaters HVH, e.g. from Websato. This device has many advantages, including low weight (approx. 2 kg), easy installation, high efficiency, etc. In this solution, PTC heaters that require the use of rare earth or lead are not used. The article should include information whether the solution proposed by the author can be an alternative to HVH electric heaters? It is worth analyzing this problem in the light of the advantages and disadvantages.
I have no major objections to the substantive part of the article, the model and results are presented in a consistent and logical manner. In my opinion, it is worth considering in Formula 8 changing the designation of electrical resistance from kp to Rp. I leave this decision to the author of the article.
The paper presents only simulation results. Does the author have a research model? It would be worthwhile to present at least preliminary measurement results in the paper and compare them with the results obtained from calculations. The measurement results would significantly increase the overall quality of the paper.
In terms of editing, the article requires correction. For example, empty lines should be removed, eg 159, 182, 212, etc. The numbering of the formulas may also require a slight correction (the numbering of the formulas should not be on the right side of the manuscript?).
Round 2
Reviewer 1 Report
Dear Author,
In the first paper version, my biggest concerns were related to complete lack of any experimental validation related to the presented numerical model. The numerical model of its own, and its results can surely be interesting, but its true value can be confirmed only by the experiments.
In the revised paper version, You have presented a preliminary experimental results which confirm validity and good accuracy of the developed numerical model. This is completely enough evidence that the numerical model can track operating parameters of a real system.
All the other elements mentioned in my review are properly addressed and involved in the paper.
Now, after revision, I have no more concerns related to this paper, and I can recommend its publication.
My congratulations to the Author.
Reviewer 3 Report
After reading the revised version of the article and reading the answers to the reviewer's questions, I accept the article for publication in the mdpi appl sci journal.